# The association between dietary patterns before and in early pregnancy and the risk of gestational diabetes mellitus (GDM): Data from the Malaysian SECOST cohort

**Heng Yaw Yong** [1], **Zalilah Mohd Shariff**[1] *, **Barakatun-Nisak Mohd Yusof**[1], **Zulida Rejali**[2], **Geeta Appannah**[1], **Jacques Bindels**[3], **Yvonne Yee Siang Tee**[4], **Eline M. van der Beek**[3,5]

**1** Department of Nutrition and Dietetics, Faculty of Medicine and Health Sciences, Universiti Putra Malaysia, Selangor, Malaysia, **2** Department of Obstetrics and Gynaecology, Faculty of Medicine and Health Sciences, Universiti Putra Malaysia, Selangor, Malaysia, **3** Danone Nutricia Research, Utrecht, The Netherlands, **4** Danone Dumex (M) Shd Bhd, Nilai, Negeri Sembilan, Malaysia, **5** Department of Pediatrics, University Medical Centre Groningen, University of Groningen, Groningen, The Netherlands

* zalilahms@upm.edu.my

**Data Availability Statement:** The Medical Research Ethics Committee (MREC), Ministry of Health Malaysia, has imposed on the restriction of

## Abstract

Generally, dietary patterns (DP)s have been linked to the risk of diabetes mellitus, however, only few studies examined the associations between DPs in early pregnancy and the risk of gestational diabetes mellitus (GDM). This study aims to determine the association between DPs before and during pregnancy and risk of GDM in Malaysian pregnant women. DPs were derived using principal component analysis of consumed 126 food and beverage items assessed using a validated semi-quantitative food frequency questionnaire collecting data retrospectively for pre-pregnancy, but prospectively for the first and second trimester. Three different DPs were identified at each time point and labelled as DP 1–3 (pre-pregnancy), DP 4–6 (first trimester), and DP 7–9 (second trimester). About 10.6% (n = 48) of pregnant women were diagnosed with GDM in our cohort. Women with high adherence (HA) to DP 2 (adjusted OR: 0.45, 95% CI: 0.20–0.91) and DP 5 (adjusted OR: 0.28, 95% CI: 0.11–0.68) showed a significantly reduced risk of GDM compared to women with low adherence (LA). Other DPs were not significantly associated with GDM risk. Compared to women with GDM, non-GDM women showed HA scores for all DPs throughout pregnancy. Overall, a relative low percentage of women with GDM was found in this cohort. The risk was lower in women with HA to a relatively unhealthy dietary pattern, i.e. DP 2 and DP 5. The lower body mass index (BMI) status and energy intake of women showing a HA to DP 2 in the first trimester may underlie the observed association with a lower GDM risk. Additionally, genetic variance might explain the less susceptibility to GDM despite HA to unhealthy DPs among non-GDM women.

disclosure of data contain potentially identifying patient information. However, the data analyzed in the current study will be available by contacting the MREC at nmrr@nmrr.gov.my or the SECOST project leader, Prof Dr. Zalilah Mohd Shariff (zalilahms@upm.edu.my).

**Funding:** SECOST was supported by a research grant from Danone Dumex (Malaysia) Shd. Bhd. The funder had no role in study design, data collection and analysis, decision to publish, or preparation of the manuscript.

**Competing interests:** Jacques Bindels and Eline van der Beek are employees of Danone Nutricia Research and Yvonne Yee Siang Tee of Danone Dumex Malaysia. The authors declare that they have no conflict of interest. This does not alter our adherence to PLOS ONE policies on sharing data and materials.

## Introduction

Gestational Diabetes Mellitus (GDM), an increasingly common type of hyperglycemia during pregnancy, follows the increasing trends of obesity and Type 2 Diabetes Mellitus (T2DM). In 2017, the International Diabetes Federation (IDF) estimated that 20.9 million (16.2%) of live births were affected by hyperglycemia in pregnancy, and about 85.1% of these live births were due to GDM [1]. The National Health and Morbidity Survey (2016) reported that the prevalence of GDM among Malaysian mothers aged 15–49 years old was 13.5% [2]. The GDM rate in Malaysian population (8.7–29.7%) [3–5] was significantly higher than the reported rates in many Western (2.0–9.2%) [6,7] and Asian countries (2.8–25.0%) [8–11].

Dietary intake during pregnancy is commonly assessed through the intakes of energy, macronutrients, micronutrients or food groups [12,13], which may then be examined in relation to an imbalanced maternal diet and poor maternal nutritional status. Dietary pattern is a relatively new approach that describes a combination of commonly consumed foods [14] that allows for the diet to be described as a whole [15]. Nutritional health outcomes are often the result of multiple synergies among nutrients and foods rather than just the sum of the individual food [16]. Although there are several approaches to identifying dietary patterns of pregnant women, the posterior-approach derived from principal component analysis (PCA) is the most commonly used for deriving dietary patterns during pregnancy [17–20]. PCA is a technique to reduce a large of correlated variables into a smaller number of components [21,22], revealing the underlying structure within diets of the population. Numerous studies examining the role of PCA derived DPs in adverse pregnancy outcomes, such as GDM, pre-eclampsia, and preterm birth [23–25] have been published.

Associations between DP before and during pregnancy with risk of GDM has been examined [26–30]. Most studies showed that an adoption of the prudent pattern, diet that was rich in vegetables, fruits, whole grains and legumes showed significant lower risk of GDM [26–28], whereas adhering to the Western pattern, which is characterized by high intakes of red meat, processed meats and refined foods, was associated with increased risk of GDM [23,26,31]. There are several suggested mechanisms by which Western pattern may increase the risk of GDM through inflammation, and placental dysfunction [26,29,30].

Despite the numerous studies on the association between DPs and risk of GDM, findings are mostly inconsistent. These inconsistencies could be due to different study designs and diagnostic criteria of GDM, methods to measure dietary intake (e.g. diet recall, diet record, food frequency questionnaire), and socio-demographic factors (e.g. age, ethnicity, and BMI). It could also be attributed to the different food items reported for the respective DPs. For example, the "prudent" DP, which is a diet delivers health benefits, observed in a prospective cohort study of women in Northern China was characterized by higher intakes of dark-colored vegetables and deep-sea fish [26], while the "prudent" DP of women in the Born in Guangzhou Cohort Study (BIGCS) was predominantly dairy products, nuts, eggs, fish, soups, and fruits [27].

Previous studies on DPs and GDM have focused on assessment of such relationship at a single time point, either pre-pregnancy [29,30,32–34] or during pregnancy, before diagnosis of GDM [28,35–37]. Most studies on DPs were limited to Western countries [28,35–37], with only two studies conducted in Asian populations [27,38]. As there are considerable differences in diet and lifestyle behaviors, the DPs of Malaysian pregnant women could be different from those reported in Western as well as other Asian populations. Most people eat a combination of healthy and less healthy foods, but the impact of mixed dietary patterns on the risk of GDM remains unknown. Thus, the objectives of this prospective study are to describe the DPs before and during pregnancy in a sample of Malaysian pregnant women and to determine the association between DPs and the risk of GDM.

## Materials and methods

### Study design and location

The SECOST (Seremban Cohort Study) project is an on-going prospective study in which pregnant women are followed-up up to 1 year postpartum, and their infants are followed-up every 6 months until 2 years old. Women in the first trimester (8 – 10th weeks of gestation) of pregnancy were recruited from three maternal and child health (MCH) clinics in Seremban District, Negeri Sembilan, Malaysia. Detailed descriptions of the study methodology have been previously published [39], and only a brief overview is provided here. All pregnant women were eligible to participate unless they had one or more exclusion criteria as published previously.

A total of 737 pregnant women were enrolled in the SECOST study. About 22.7% (n = 167) and 16.0% (n = 118) women were excluded at first and second visit, respectively (Fig 1). The present study reported on data of 452 pregnant women only. The adequacy of the sample size of pregnant women was checked to ensure a 5% statistical significance level and 80% of power.

The study protocol was approved by the Medical Research Ethics Committee (MREC), Universiti Putra Malaysia (UPM/FPSK/100-9/2-MJKEtika), and the Medical Research Ethics Committee (MREC), Ministry of Health Malaysia (KKM/NIHSEC/08/0804/P12-613). Permission to conduct this study was also obtained from the Seremban District Health Office. All women provided written informed consent and a study manual.

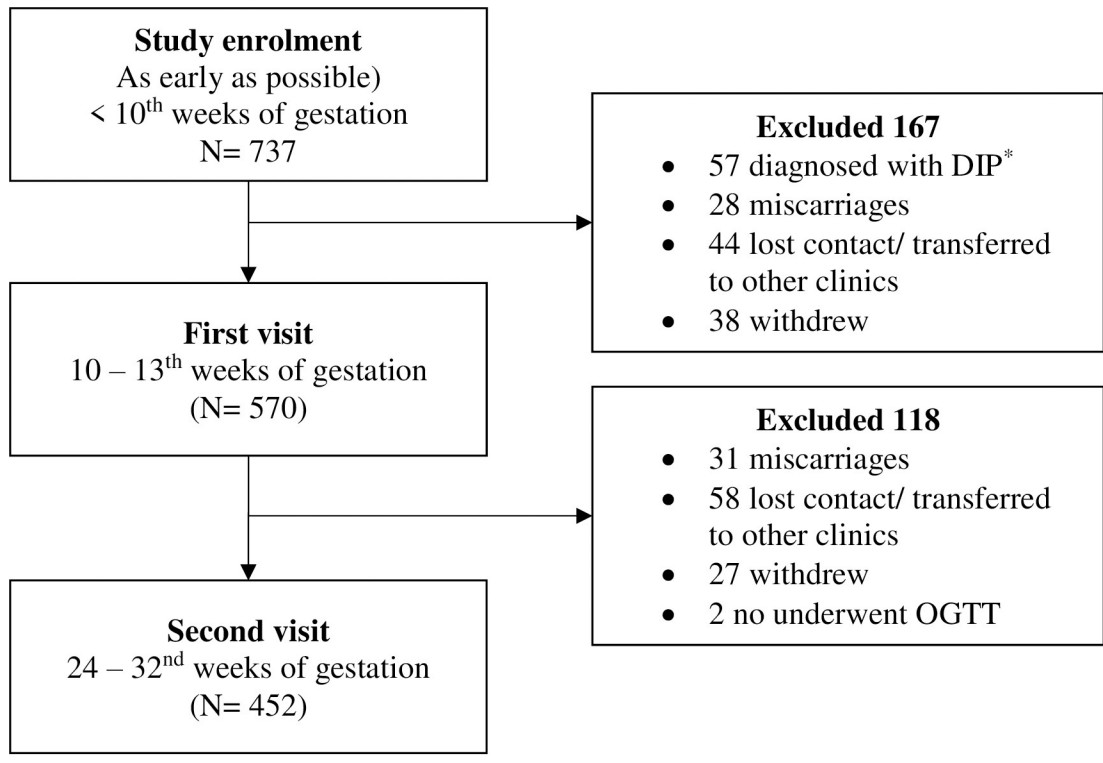

\* Diabetes in pregnancy (DIP) was diagnosed as either or both FGP > 5.6 mmol/L or 2hPG > 7.8 mmol/l (MOH, 2015).

**Fig 1. Flowchart of study respondents.**

## Dietary patterns (DPs)

A validated 126-food item semi-quantitative Food Frequency Questionnaire (SFFQ) was utilized to assess food consumption patterns [40]. The dietary assessments were conducted at three time points: at the first prenatal visit (9.82 ± 2.51 weeks of gestation to collect data on DPs before pregnancy), the first trimester (12.26 ± 1.58 weeks of gestation) and the second trimester (26.73 ± 1.64 weeks of gestation), respectively. The dietary questionnaire used for this study has shown a good relative reliability and validity of food intakes of Malaysian pregnant women [41]. The food items were categorized into 17 food groups based on similar nutrient characteristics and considering similar in culinary usage [42]. They include rice, noodles & pasta; bread, cereal & cereal products; poultry & meat; processed meat; fish & seafood; eggs; nuts, seeds & legumes; milk & dairy products; green leafy vegetables; other vegetables; fruits; tea and coffee; high energy beverages; sweet foods, sugar, spread & creamer, condiments & spices, and oils & fats.

Principal component analysis (PCA) was performed based on 17 food and beverage items to derive dietary patterns. PCA was performed separately for each time point and the factors were rotated by an orthogonal transformation (varimax rotation) to maintain uncorrelated factors and greater interpretability. The number of factors was determined by eigen values greater than 1.5 [43]. The data suitability was assessed before performing PCA. Inspection of the correlation matrix on the food groups revealed the presence of several correlation coefficients of 0.2 and above. Food groups with absolute factor loading with values < 0.30 were not listed in the final model. The Kaiser-Meyer-Olkin measure of sampling adequacy was used to ensure that the sample size was adequate for these analyses. The Kaiser-Meyer-Oklin values were 0.68 (pre-pregnancy), 0.71 (first trimester), and 0.65 (second trimester). Three characteristic DPs were identified at each time point and labelled as DP 1–3 (pre-pregnancy), DP 4–6 (first trimester), and DP 7–9 (second trimester). The factor scores for each DP and for each individual were calculated by summing the food item intakes weighted by their factor loadings [44]. The factor score was then categorized into tertiles (1st tertile = low adherence (LA), 2nd tertile = moderate adherence (MA), and 3rd tertile = high adherence (HA)) for ease of interpretation in the subsequent analysis. More detailed description of DPs before and during pregnancy was published elsewhere (DOI: 10.4162/nrp.2019.13.3.230) [42].

## Gestational diabetes mellitus (GDM)

Women with pre-existing diabetes mellitus (DM) (fasting plasma glucose (FPG) >7.0 mmol/l), or abnormal glycaemia (FPG <3.0 mmol/l or FPG >6.0 mmol/l) at study enrolment were excluded. A standard two-point diagnostic 75g oral glucose tolerance test (OGTT) was performed at 28 – 32nd weeks of gestation. A 2-ml fasting venous blood was drawn by a clinic staff nurse before ingestion of a standard glucose solution to obtain fasting plasma glucose (FPG). Another 2 ml of venous blood was drawn at 2-hours after ingestion of standard glucose solution. All blood samples were sent for analysis on the same day to determine FPG and 2-hr plasma glucose (2hPG) concentration. GDM was diagnosed if either or both FPG was ≥ 5.6 mmol/l or 2hPG is ≥ 7.8mmol/l according to Ministry of Health Malaysia guideline [45].

## Other variables

Socio-demographic information included age, education level, ethnicity, occupation status, and monthly household income. Obstetrical information (e.g. gravidity, parity, medical history GDM, and family history of diabetes mellitus) was obtained from medical records. Height was measured at study enrolment and was further categorized into tertiles based on the overall distribution, while weight was measured at each study visit using a standard instrument (SECA

digital weighing scale and SECA body meter) and standard procedures. Women were requested to recall pre-pregnancy body weight. Pre-pregnancy body mass index (BMI) (kg/m$^2$) was calculated as pre-pregnancy weight divided by the square of height and classified according to the recommendation of World Health Organization [46].

### Statistical analysis

All analyses were performed using SPPS version 24. Exploratory Data Analysis (EDA) was carried out to determine the normality and homogeneity of the data. All continuous variables were normally distributed. Therefore, no transformation was performed. Basic descriptive statistics were generated such as means and standard deviations for the continuous variables, while for categorical variables, frequency, and percentage distribution. Chi-square test of independence or Fisher's exact test and Independent t-test were used to assess the association between women characteristics with risk of GDM, respectively for continuous and categorical variables.

Generalized Linear Mixed Model (GLMM) was performed to assess the association between tertile of adherence to DPs and the risk of GDM, with MCH clinics and gestational week at OGTT as random effects. Potential covariates included in the multivariable model were age (continuous), ethnicity (Malays vs Non-Malays), medical history of GDM (yes vs no) and family history of DM (yes vs no). The lowest tertile of adherence was set as the reference group. Adjusted odds ratio (OR) with 95% confidence interval (CI) of the association between DPs and GDM were presented. A further stratified analysis was performed to determine the associations between DP 2 and DP 5 with GDM by pre-pregnancy BMI. Repeated measures ANOVA was used to plot the DP trajectory for GDM and non-GDM groups. Three major patterns were identified at each time point: pattern 1 (DP 1, DP 4, and DP 7), pattern 2 (DP 2, DP 5, and DP 8), and pattern 3 (DP 3, DP 6 and DP 9). Sensitivity analyses were run between acceptable reporters and under-reporters of energy intake. The findings did not change the main findings (S1 Table). The cut off point for statistical significance was set at p<0.05.

### Results

Table 1 compares the characteristics of 404 non-GDM and 48 GDM women. Age was similar in both groups, with the mean of 30.04 ± 4.50 years in non-GDM and 29.80 ± 4.39 years in GDM women. Majority of the women were Malay (88.9–89.6%), had secondary and lower education level (41.7–46.6%), currently employed (68.6–68.7%), and had low monthly household income (58.3–63.4%). Women with GDM had a significantly higher percentage of history of GDM (16.7%) and family history of diabetes mellitus (37.5%) compared to non-GDM women (history of GDM = 5.7%; family history of diabetes mellitus = 23.0%). Most of the women had a height below 1.55 m (37.2–45.8%) and normal pre-pregnancy BMI (47.9–56.2%). We observed no significant differences in height and pre-pregnancy BMI between non-GDM (mean height = 1.56 ± 0.06 m; mean pre-pregnancy BMI = 23.64 ± 4.81 kg/m$^2$) and GDM women (mean height = 1.56 ± 0.06 m; mean pre-pregnancy BMI = 24.45 ± 4.75kg/m$^2$).

Table 2 presents the absolute factor loadings ≥ 0.30 for each DP. Three specific DPs were observed at each time period before and during pregnancy: pre-pregnancy (DP 1–3), the first trimester (DP 4–6), and the second trimester (DP 7–9). DP 1, DP 4 and DP 7 were considered as the 'prudent, healthy pattern', which was predominantly plant-based with high factor loadings for other vegetables, nuts, seeds & legumes, green leafy vegetables, fruits (DP 1 and DP 4), eggs (DP 1), and rice, noodles & pasta (DP 7). DP 2, DP 5 and DP 8 had high factor loadings for condiments & spices, and sugar, spread & creamer, with DP 5 having additional oils & fats. This pattern could be considered as a DP compliant to the local food culture. DP 3, DP 6 and

**Table 1. Characteristics of women (N = 452).**

| | Maternal glycemia[a] | | p-value[b] |
|---|---|---|---|
| | **Non-GDM (n = 404)** | **GDM (n = 48)** | |
| Age (years) | | | |
| • < 35 | 340 (84.2) | 39 (81.3) | 0.61 |
| • ≥ 35 | 64 (15.8) | 9 (18.7) | |
| Mean ± SD | 30.04 ± 4.50 | 29.80 ± 4.39 | 0.73 |
| Ethnicity | | | |
| • Malay | 359 (88.9) | 43 (89.6) | 0.88 |
| • Non-Malay | 45 (11.1) | 5 (10.4) | |
| Education level (years) | | | |
| • Secondary and lower | 188 (46.6) | 20 (41.7) | 0.57 |
| • STPM/ Matriculation/ Diploma/ Certificate[c] | 133 (32.9) | 15 (31.3) | |
| • Tertiary and above | 83 (20.5) | 13 (27.0) | |
| Mean ± SD | 12.93 ± 3.53 | 13.21 ± 2.70 | 0.45 |
| Occupation status | | | |
| • Unemployed | 127 (31.4) | 15 (31.3) | 0.98 |
| • Employed | 277 (68.6) | 33 (68.7) | |
| Monthly household income (RM)[d] | | | |
| • Low (< 3860) | 256 (63.4) | 28 (58.3) | 0.08 |
| • Middle (3860–8319) | 138 (34.1) | 16 (33.3) | |
| • High (≥ 8320) | 10 (2.5) | 4 (8.4) | |
| Mean ± SD | 3683.63 ± 2015.56 | 4089.58 ± 2319.50 | 0.20 |
| Parity | | | |
| • Nulliparous | 145 (35.8) | 15 (31.3) | 0.73 |
| • Primiparous | 121 (30.0) | 14 (29.2) | |
| • Multiparous | 138 (34.2) | 19 (39.5) | |
| Medical history | | | |
| • GDM | 23 (5.7) | 8 (16.7) | 0.01* |
| Family history | | | |
| • Diabetes mellitus | 93 (23.0) | 18 (37.5) | 0.03* |
| Height (m) | | | |
| • < 1.55 | 150 (37.2) | 22 (45.8) | 0.33 |
| • 1.55–1.58 | 123 (30.4) | 10 (20.8) | |
| • ≥ 1.59 | 131 (32.4) | 16 (33.4) | |
| Mean ± SD | 1.56 ± 0.06 | 1.56 ± 0.06 | 0.34 |
| Pre-pregnancy BMI (kg/m$^2$) | 23.64 ± 4.81 | 24.45 ± 4.75 | 0.27 |
| • Underweight (< 18.5) | 44 (10.9) | 4 (8.3) | 0.47 |
| • Normal (18.5–24.9) | 227 (56.2) | 23 (47.9) | |
| • Overweight (25.0–29.9) | 88 (21.8) | 15 (31.3) | |
| • Obese (≥ 30.0) | 45 (11.1) | 6 (12.5) | |

Note.

[a]GDM was classified according to Ministry of Health Malaysia criteria as either or both FPG ≥ 5.6 mmol/l or 2-hour plasma glucose ≥ 7.8 mmol/l.

[b]p-value for differences between GDM and non-GDM groups were examined by independent t-test for continuous variables and chi-square for categorical variables.

[c]STPM–Malaysian Higher School Certificate

[d]Economic Planning Unit, Prime Minister's Department, 2014.

*p<0.05

**Table 2. Dietary pattern from pre-pregnancy to second trimester (n = 452).**

| Food Groups | Pre-pregnancy | | | First trimester | | | Second trimester | | |
|---|---|---|---|---|---|---|---|---|---|
| | DP 1 | DP 2 | DP 3 | DP 4 | DP 5 | DP 6 | DP 7 | DP 8 | DP 9 |
| Other vegetables | 0.76 | | | 0.81 | | | 0.75 | | |
| Nuts, seeds & legumes | 0.65 | | | 0.30 | | | 0.41 | | |
| Green leafy vegetables | 0.65 | | | 0.78 | | | 0.83 | | |
| Fruits | 0.48 | | | 0.64 | | | | | 0.56 |
| Eggs | 0.46 | | | | | 0.65 | | | 0.56 |
| Milk & dairy products | 0.35 | | | | | 0.48 | | | 0.52 |
| Sugar, spread & creamer | | 0.98 | | | 0.97 | | | 0.97 | |
| Condiments & spices | | 0.98 | | | 0.97 | | | 0.97 | |
| Rice, noodles & pasta | | | 0.74 | | | 0.42 | 0.40 | | |
| Oils & fats | | | 0.67 | | 0.36 | | | | |
| High energy beverages | | | 0.58 | | | 0.45 | | | 0.54 |
| Fish & seafood | | | 0.46 | | | 0.44 | | | 0.61 |
| Sweet foods | | | 0.34 | | | 0.39 | | | 0.37 |
| Poultry & meat | | | 0.33 | | | 0.28 | | | 0.41 |
| Bread, cereal & cereal products | | | | | | 0.66 | | | 0.64 |
| Processed meat | | | | | | 0.41 | | | 0.42 |
| **Total variance** | 12.89% | 12.58% | 12.04% | 11.93% | 12.78% | 13.86% | 10.12% | 12.67% | 16.31% |

Only food groups with absolute factor loadings > 0.30 were retained in each pattern for simplicity.

DP 9 were characterized by high protein (poultry, meat, processed meat, dairy, egg and fish), sugars (mainly as high energy beverages, and sweet foods), and energy (bread, cereal & cereal products, rice, noodles & pasta), and only for DP 9 with additional fruits This pattern represented a combination of various food groups, but may be more equivalent to a Western unhealthy diet.

Table 3 shows the odds ratio (ORs) and 95% CI for the risk of GDM according to adherence tertiles to DPs before and during pregnancy. During pre-pregnancy, a similar trend was observed for all DPs, in which women with HA were at reduced risk of GDM. In the first trimester, women with HA to DP 4 and DP 5 were at reduced risk of GDM, and this trend remained similar in the second trimester (DP 7 and DP 8). Meanwhile, women with HA to DP 6 were at higher risk of GDM in the first trimester, but they were at lower risk for GDM in the second trimester (DP 9). The associations between adherence to a DP and the risk of GDM were generally not significant, except for DP 2 and DP 5, e.g. women with HA to DP 2 (adjusted OR: 0.45, 95% CI: 0.20–0.91) and DP 5 (adjusted OR: 0.28, 95% CI: 0.11–0.68) had significantly reduced risk of GDM compared to women with LA. The findings of the stratified analysis showed that the significant association between DP 5 and GDM was only observed among underweight or normal-weight women. Meanwhile, a significant association between DP 2 and GDM risk was found only among obese women (Table 4).

DP trajectory from pre-pregnancy to second trimester of pregnancy for non-GDM and GDM women are depicted in Fig 2. The number of women with GDM (n = 48) were relatively small. There were significant differences in pattern 1, 2 and 3 between non-GDM and GDM women. While non-GDM women maintained high adherence to all three DPs before and during pregnancy, GDM women changed their diets during pregnancy. For pattern 1 (DP 1, DP 4 and DP 7 –prudent diet, high in fruits, vegetables, nuts, seeds, legumes, eggs, milk and dairy), GDM women had low DP score before pregnancy, decreased score at the first trimester and

**Table 3. Adjusted odd ratios and 95% confidence intervals for the association between dietary pattern and GDM (N = 452).**

| Dietary pattern [b] | Maternal glycemia [a] | |
|---|---|---|
| | **Adjusted OR** | **p-value** |
| **Pre-pregnancy** | | |
| DP 1 | | |
| • LA | 1.00 | |
| • MA | 0.83 [0.39–1.76] | 0.62 |
| • HA | 0.82 [0.38–1.75] | 0.60 |
| DP 2 | | |
| • LA | 1.00 | |
| • MA | 0.65 [0.31–1.37] | 0.25 |
| • HA | **0.45 [0.20–0.91]** | **0.04**[*] |
| DP 3 | | |
| • LA | 1.00 | |
| • MA | 0.70 [0.33–1.49] | 0.36 |
| • HA | 0.79 [0.38–1.64] | 0.52 |
| **First trimester** | | |
| DP 4 | | |
| • LA | 1.00 | |
| • MA | 0.78 [0.37–1.63] | 0.51 |
| • HA | 0.81 [0.38–1.71] | 0.57 |
| DP 5 | | |
| • LA | 1.00 | |
| • MA | 0.70 [0.35–1.40] | 0.31 |
| • HA | **0.28 [0.11–0.68]** | **0.01**[*] |
| DP 6 | | |
| • LA | 1.00 | |
| • MA | 1.27 [0.59–2.72] | 0.54 |
| • HA | 1.15 [0.54–2.47] | 0.72 |
| **Second trimester** | | |
| DP 7 | | |
| • LA | 1.00 | |
| • MA | 0.67 [0.33–1.38] | 0.17 |
| • HA | 0.51 [0.24–1.11] | 0.07 |
| DP 8 | | |
| • LA | 1.00 | |
| • MA | 0.69 [0.32–1.50] | 0.35 |
| • HA | 0.73 [0.34–1.58] | 0.42 |
| DP 9 | | |
| • LA | 1.00 | |
| • MA | 1.24 [0.60–2.55] | 0.56 |
| • HA | 0.73 [0.33–1.63] | 0.44 |

Note.

[a] The reference category is non GDM.

[b] Dietary patterns were classified in tertiles of adherence (1st tertile = low adherence (LA); 2nd tertile = moderate adherence (MA) & 3rd tertile = high adherence (HA)).

Adjusted for clinic, gestational week at OGTT performed, maternal age, ethnicity, medical history of GDM and family history of DM.

[*]$p < 0.05$

**Table 4. Adjusted odd ratios and 95% confidence intervals for the associations between DP 2 and DP 5 with GDM stratified by pre-pregnancy BMI (n = 452).**

| Pre-pregnancy BMI | Dietary pattern [b] | Maternal glycemia [a] | |
| --- | --- | --- | --- |
| | | Adjusted OR | p-value |
| **Underweight and normal weight (n = 298)** | DP 2 | | |
| | • LA | 1.00 | |
| | • MA | 0.69 [0.25–1.90] | 0.47 |
| | • HA | 0.56 [0.20–1.58] | 0.27 |
| | DP 5 | | |
| | • LA | 1.00 | |
| | • MA | 0.71 [0.28–1.81] | 0.47 |
| | • HA | **0.27 [0.08–0.87]** | **0.02**[*] |
| **Overweight (n = 103)** | DP 2 | | |
| | • LA | 1.00 | |
| | • MA | 0.61 [0.26–1.71] | 0.34 |
| | • HA | 0.36 [0.06–1.38] | 0.08 |
| | DP 5 | | |
| | • LA | 1.00 | |
| | • MA | 0.52 [0.20–1.36] | 0.42 |
| | • HA | 0.41 [0.15–1.78] | 0.24 |
| **Obese (n = 51)** | DP 2 | | |
| | • LA | 1.00 | |
| | • MA | - | |
| | • HA | **0.42 [0.03–0.57]** | **0.04**[*] |
| | DP 5 | | |
| | • LA | 1.00 | |
| | • MA | - | |
| | • HA | 0.24 [0.08–4.61] | 0.34 |

Note.

a The reference category is non GDM.

b Dietary patterns were classified in tertiles of adherence (1st tertile = low adherence (LA); 2nd tertile = moderate adherence (MA) & 3rd tertile = high adherence (HA)).

Adjusted for clinic, gestational week at OGTT performed, maternal age, ethnicity, medical history of GDM and family history of DM.

[*]p<0.05

further decreased score at second trimester. For pattern 2 (DP 2, DP 5 and DP 8 –high in condiments & spices, and sugar, spread & creamer), a V-shaped trend was observed in GDM women, where DP score was at the lowest point in the first trimester and higher at the second trimester. For pattern 3 (DP 3, DP 6 and DP 9 –high protein, sugar and energy), GDM women showed an inverse V-shaped trend in that they had low DP score before pregnancy, the highest DP score at the first trimester and the lowest DP score at the second trimester.

## Discussion

In this prospective cohort of 452 pregnant women, three major DPs were identified before and during pregnancy. Only DP 2 and DP 5 showed significantly a lower risk of GDM after the adjustment for covariates. The study findings should be interpreted with caution as this pattern was relatively higher in loadings for sugar, spread & creamer, sauces, condiments & spices,

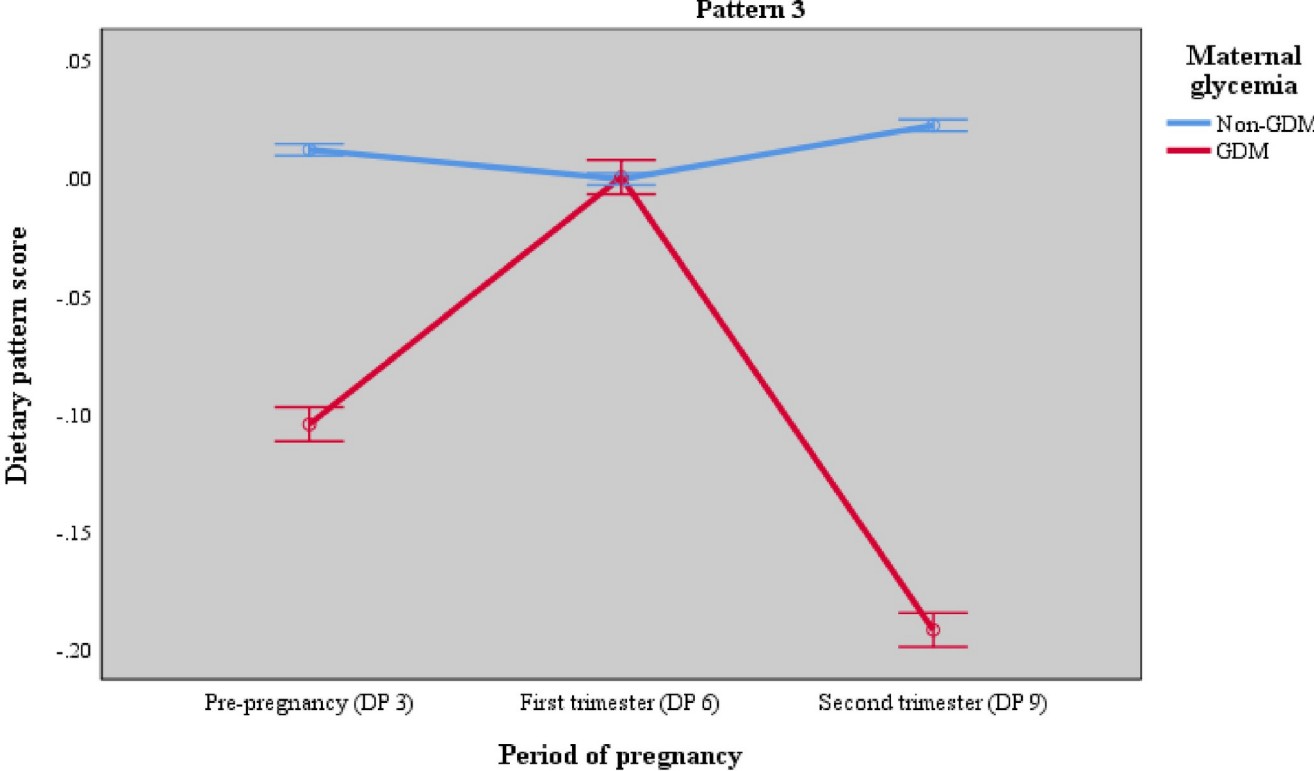

**Fig 2. DP trajectory before and during pregnancy by maternal glycemia adjusted for pre-pregnancy BMI and rate of gestational weight gain.**

and oils & fats. The percentage of under-reporting was 25.9–31.2%, calculated using the Goldberg method (rEI:BMR) with implausible reporters had rEI:BMR values that differed from physical activity levels by more than ± 2 standard deviations [47,48]. The food items observed in this pattern were mainly cooking ingredients or additions to food items consumed at main meals or snacks. However, it can be referred to as the common food pattern, as these foods are commonly consumed by the Malaysian population. This DP has not been previously identified in the Western dietary pattern [27–29,36] but was similar to the 'less-healthy' pattern (condiments, sugar, oils and fats, sweets and desserts, and tea and coffee) reported by pregnant women in Universiti Sains Malaysia (USM) Birth Cohort [49]. Thus, this pattern is believed to be the true DP of our study population. The finding also reflects specific differences in dietary patterns between the Malaysia and Western population.

Generally, women with diets high in sugar and fat show a higher risk of GDM [31,50]. Yet, this study showed that women with HA to DP 2 (characterized as high sugar and fat) in pre-pregnancy and DP 5 in the first trimester were in fact significantly at reduced risk of GDM compared to women with LA to such a relatively unhealthy diet. Subsequently, a stratified analysis was performed to determine the associations between DP 2 and DP 5 with GDM by pre-pregnancy BMI (Table 4). This analysis showed that the significant association between DP 5 and GDM was only observed among underweight or normal weight women. Meanwhile, a significant association between DP 2 and GDM risk was found only among obese women. In addition, the present study also found that among women with HA to DP 5, underweight or

normal weight women had significantly lower total energy intake in the first trimester (1,541 kcal/day) than overweight or obese women (1,624 kcal/day) (S2 Table). The mean total energy intake was also below the requirement for pregnant women in first trimester (1,690 kcal/day) [51]. Based on these data we speculate that the lower total energy intake observed in women with this dietary pattern might be protective of insulin resistance and consequently hypergly-cemia risk. Thus, a lower BMI status and energy intake of women with high HA to DP 5 may underlie the observed association with a lower GDM risk. We cannot examine the association between pre-pregnancy energy intake and DP 2 with risk of GDM as the pre-pregnancy energy intake cannot be assessed retrospectively. Nevertheless, further investigation is needed to con-firm the observed associations for underweight or normal weight women and for overweight or obese women and understand the drivers for GDM risk.

The findings from the ANOVA repeated measures showed that non-GDM women tend to have consistently higher DP scores for all three patterns before pregnancy up to the second tri-mester while GDM women showed lower and/or inconsistent DP scores from pre-pregnancy up to the second trimester of pregnancy. These trends in DP scores were similar regardless of adjustments for covariates like gestational weight gain and pre-pregnancy BMI. Pattern 2 and pattern 3 can be considered as the "less healthy" patterns due to their higher energy, fat, and sugar content. Yet, pattern 2 represents the typical Malaysian DP in which additional cooking oils, condiments and sugary substances are used to prepare standard fresh food items. Previous studies have shown that genetic factors are important in determining susceptibility to GDM [52,53]. Inherited abnormalities of pancreatic islet β-cell function and/ or β-cell mass may be implicated in the etiology of GDM [54]. Thus, it is plausible that non-GDM women in the present study have genetic variants that make them less susceptible to the risk of GDM and can explain why they are not at risk of GDM despite having also high DP scores for less healthy DPs. In general, the percentage of women that developed GDM in this cohort was low, also compared to the reported prevalence of GDM in Malaysia [2,3,5]. This may indicate that the SECOST study was conducted among a relatively traditional and healthy population. Further study is needed to identify the factors including genetic that explain susceptibility for GDM and examine the associations between genes, DPs, and BMI with the risk of GDM.

Similar to previous studies [28,29,33], the present study supported that only DPs in the first trimester and before pregnancy were significantly associated with the risk of GDM. Maternal metabolism in early pregnancy is critical not only for maternal health but also gestational pro-gramming [55,56]. During the first few weeks of pregnancy, the presence of placenta causes a reduction in growth hormone levels, resulting in enhanced insulin sensitivity [57]. GDM occurs if β-cells are unable to compensate for the changes in insulin resistance [58]. In another word, women are exposed to metabolic dysregulation in early pregnancy, and this metabolic abnormality may increase the risk of adverse pregnancy outcomes. Oostvogels et al. (2017) showed that pre-pregnancy maternal weight status and early pregnancy maternal lipid profile are independently associated with offspring adiposity [59], which reinforces the important role of maternal metabolism before and in early pregnancy. The precise underlying mecha-nisms between pre-pregnancy and early pregnancy DPs with the risk of GDM, however, remains to be elucidated. Nevertheless, the available evidence seems to support that women planning for pregnancy or in early pregnancy should be the target of lifestyle intervention for preventing GDM [60–62].

This study is not without any limitation. The results of the PCA approach might be affected by several arbitrary decisions such as determination of a number of factors to extract, the com-ponents labeling, the method of rotation, and even their interpretation [63]. In this study, the total variance of 3 patterns explained in each time point is relatively low (<40%). However, this finding was comparable with other previous studies that reported a total variance of 16.6–

32.9% [23,27,31]. Besides, there is also the possibility of recall bias and misreporting when using SFFQ for dietary assessment. Nevertheless, the use of well-trained enumerators in data collection could reduce these errors. Although the effects of potential covariates were adjusted in the statistical methods, residual confounding of unknown confounders cannot be excluded. Besides, nutrition knowledge for study population is unknown. Future studies could include extending confounders to the model as well as to determine the nutrition knowledge of pregnant women. Despite these limitations, the present study is worthwhile as it provides important insights into the association between DPs before and during pregnancy and the risk of GDM, which such information is lacking in the literature.

## Conclusions

The present study found that diets rich in sugar, spread and creamer, spices and condiments (DP 2 and DP 5) were significantly associated with reduced risk of GDM. The unexpected findings could be due to factors such as, lower BMI, and reduced energy intake. Furthermore, non-GDM women maintained a high DP score for all DPs from pre-pregnancy until second trimester of pregnancy, meanwhile GDM women showed lower and/or inconsistent DP scores, suggesting that non-GDM might have a lower genetic susceptibility to GDM as they are not at risk of GDM despite having also high DP scores for less healthy DPs (pattern 2 and pattern 3). A future large-scale prospective study or a well-designed randomized controlled trial is warranted to further confirm the association between the dietary pattern and the GDM in the Malaysian population. Such findings could inform efforts to strengthen existing or develop appropriate health and nutrition strategies to address the increasing rate of GDM in Malaysia.

## Supporting information

**S1 Table. Adjusted odd ratios and 95% confidence intervals for dietary pattern 2 and GDM (n = 452) [Sensitivity analysis].**
(DOCX)

**S2 Table. Energy intake in first trimester of women by pre-pregnancy BMI.**
(DOCX)

## Acknowledgments

We are extremely grateful to all the women who took part in this study, the medical officers and nurses for their assistance during data collection, and the whole SECOST team, which includes interviewers, research assistants, and volunteers.

## Author Contributions

**Conceptualization:** Heng Yaw Yong, Zalilah Mohd Shariff.

**Formal analysis:** Heng Yaw Yong.

**Funding acquisition:** Zalilah Mohd Shariff.

**Methodology:** Heng Yaw Yong, Barakatun-Nisak Mohd Yusof.

**Project administration:** Heng Yaw Yong.

**Supervision:** Heng Yaw Yong, Zalilah Mohd Shariff, Zulida Rejali, Geeta Appannah, Eline M. van der Beek.

**Writing – original draft:** Heng Yaw Yong, Zalilah Mohd Shariff.

**Writing – review & editing:** Heng Yaw Yong, Zalilah Mohd Shariff, Barakatun-Nisak Mohd Yusof, Zulida Rejali, Geeta Appannah, Jacques Bindels, Yvonne Yee Siang Tee, Eline M. van der Beek.

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
