## [Decision Letter · Decision Letter 0]

18 Sep 2019

PONE-D-19-22261

The association between dietary patterns before and in early pregnancy and the risk of gestational diabetes mellitus (GDM): data from the Malaysian SECOST cohort

PLOS ONE

Dear Dr Yong,

Thank you for submitting your manuscript to PLOS ONE. After careful consideration, we feel that it has merit but does not fully meet PLOS ONE’s publication criteria as it currently stands. Therefore, we invite you to submit a revised version of the manuscript that addresses the points raised during the review process.

• The justification for using the Principal Component Analysis is not clear because it is possible to assess the relationship between frequency of consumption of specific food groups (e.g. 17 food groups mentioned in the article) and risk of Gestational Diabetes Miletus (GDM) directely without extracting hidden DPs. The identification of hidden dietary patterns (principal factors/components) through the PCA simply complicated the interpretation of the findings.

• The authors presented conflicting findings about the significance of diet and dietary pattern for GDM. At some point they stated that DP2 and DP5 generated by PCA are significant predictors of GDM while latter they argued that dietary pattern as assessed by FFQ has no association with GDM. 

• In the abstract it is stated that “Our FFQ analysis did not reveal any clear associations between (pre)pregnancy dietary patterns and the risk to develop GDM (or not) …..” however in the results section, no result is provided regarding the direct association between frequency of consumption of food groups and risk of GDM. 

I recommend the authors to focus on the relationship between frequency of consumption of specific food groups (as measured by FFQ) and occurrence of GDM. Kind of multivariable analyses for assessing the association between the 17 food groups and GDM may provide practically meaningful findings.

• Table 3 presents the results of three different models. I recommend the authors to make interpretation based on one model which they consider as the best.

• The authors did not satisfactorily explain how DPs relatively higher in loadings for sugar, spread & creamer, sauces, condiments & spices, and oils & fats are associated with reduced risk of GDM. In the discussion section, they mentioned that the significant association was only observed among underweight and normal weight women; however the results of this stratified analysis had not been present in the results section.

We would appreciate receiving your revised manuscript by Nov 02 2019 11:59PM. To enhance the reproducibility of your results, we recommend that if applicable you deposit your laboratory protocols in protocols.io, where a protocol can be assigned its own identifier (DOI) such that it can be cited independently in the future. For instructions see: http://journals.plos.org/plosone/s/submission-guidelines#loc-laboratory-protocols

We look forward to receiving your revised manuscript.

Kind regards,

Samson Gebremedhin Gebreselassie, PhD

Academic Editor

PLOS ONE

"SECOST was supported by a research grant from Danone Dumex (Malaysia) Shd. Bhd. We are extremely grateful to all the women who took part in this study, the medical officers and nurses for their assistance during data collection, and the whole SECOST team, which includes interviewers, research assistants, and volunteers."

 "No - The funder had no role in study design, data collection and analysis, decision to publish, or preparation of the manuscript."

Additionally, because some of your funding information pertains to commercial funding, we ask you to provide an updated Competing Interests statement, declaring all sources of commercial funding.

In your Competing Interests statement, please confirm that your commercial funding does not alter your adherence to PLOS ONE Editorial policies and criteria by including the following statement: "This does not alter our adherence to PLOS ONE policies on sharing data and materials.” as detailed online in our guide for authors  http://journals.plos.org/plosone/s/competing-interests.  If this statement is not true and your adherence to PLOS policies on sharing data and materials is altered, please explain how.

Please include the updated Competing Interests Statement and Funding Statement in your cover letter. We will change the online submission form on your behalf.

Additional Editor Comments (Section-by-section comments):

Background

• Please provide clear definitions for dietary pattern and “prudent diet”.

• Page 3, line 73-77: the sentences are not clear. Please revisit them.

• The background section should give a highlight of what other studies conducted in Western or Asian population have found about the relationship between dietary pattern and GDM.

Methods

• Please comment on the adequacy of the sample size (power) for addressing the research questions at hand.

• Line 184: please provide additional information on how the sensitivity analysis was made.

• Line 188-91: not clear. Please revisit it.

• Line 138-40: “The factor scores for each dietary pattern and each individual were calculated by summing intakes of food items weighted by their factor loadings”. Line 187-188: ANOVA was used to plot the DP trajectory with similar pattern by maternal glycemia categories. These sentences are not clear, please revisit them.

• “Three major patterns were identified before and during pregnancy: pattern 1 consisted of DP 1, DP 4, and DP 7, pattern 2 consisted of DP 2, DP 5, and DP 8, and pattern 3 included DP 3, DP 6 and DP 9.” The sentence is not clear.

Results

• Table 1: What was the justification for using age of 30 as a cut-off here? Use of 35 years as a cut-off would be practically meaningful b/c the risk of many obstetric complications tends to increase after the age of 35 years. Similarly, the cutoffs used for classifying height has not been justified.

• Table 1: STPM/ Matric??? Not clear

• If the information was collected during pregnancy, how did you manage to calculate pre-pregnancy BMI?

• Line 214-15: “Three specific DPs were observed at each time period before and during pregnancy”. It is not clear how you decided to extract three components at each round. Did you evaluate the scree plot or decide take all the components having Eigenvalues above 1? Or did you simply take the first three components with the highest Eigenvalues?

• Table 2: the combined total variance explained by the three principal components is relatively low (<40%) in all of the three rounds (pre-pregnancy, first trimester and second trimester). It is usually recommended that at least 60% of the variance should be explained by the extracted components. Do you think that the PCA has adequately explained the original 17 food groups?

• I could not get the results of the “sensitivity analysis” mentioned in the methods section.

• Line 259-271: the paragraph is not clear. Please revisit it again.

Discussion

• Line 297-9: “Subsequently, a stratified analysis was performed to determine the associations between DP 2 and DP 5 with GDM by pre-pregnancy BM”. Please provide the findings of the stratified analysis in the results section. I see nothing about this analysis in the methods and results section.

Reviewers' comments:

Reviewer's Responses to Questions

**Comments to the Author**

1. Is the manuscript technically sound, and do the data support the conclusions?

Reviewer #1: No

Reviewer #2: Partly

2. Has the statistical analysis been performed appropriately and rigorously? 

Reviewer #1: No

Reviewer #2: Yes

3. Have the authors made all data underlying the findings in their manuscript fully available?

Reviewer #1: Yes

Reviewer #2: Yes

4. Is the manuscript presented in an intelligible fashion and written in standard English?

Reviewer #1: No

Reviewer #2: Yes

5. Review Comments to the Author

Reviewer #1: N/A

Reviewer #2: Thank you for asking me to review this interesting paper which has results contrary to expectations. The findings suggest that an unhealthy diet is protective against the development of gestational diabetes in a Malaysian population. The work is overall well conducted and well put together but the conclusions need further explanation and discussion in order to reconcile the findings.

Women eating DP 2 and 5 had a diet rich in sugar, spread and creamer, spices and condiments, but this is unlikely to account for all calories. It would be helpful to know if these women were eating much else besides? Perhaps a 'typical day' idea of what this diet looked like would be useful. It would also be helpful to know what exactly the number of calories were for each group. If this group was eating only small amounts of other food stuffs then it is possible the calorie amount was very different.

What %carbohydrate was seen in each group? Is there information on dietary macronutrients? It is also possible that the difference in carb or fat content also drives this association.

I personally would urge caution in interpreting these findings and would suggest that the relevance of this to public health remains unclear. This is not reason to reject this paper, but there is a risk the findings could be misinterpreted.

6. PLOS authors have the option to publish the peer review history of their article (what does this mean?). If published, this will include your full peer review and any attached files.

Reviewer #1: No

Reviewer #2: No

---

## [Author Response · Author response to Decision Letter 0]

20 Nov 2019

Response to the decision letter

We are grateful to the editor and reviewers for their constructive comments on our manuscript. We have addressed all comments and suggestions accordingly and submitted a revised version of the manuscript for further consideration by the journal. Specifically, we have provided point-by-point response to comments / suggestion put forth by the editor and reviewers.

1. The justification for using the Principal Component Analysis is not clear because it is possible to assess the relationship between frequency of consumption of specific food groups (e.g. 17 food groups mentioned in the article) and risk of Gestational Diabetes Miletus (GDM) directly without extracting hidden DPs. The identification of hidden dietary patterns (principal factors/components) through the PCA simply complicated the interpretation of the findings.

Response:

A paragraph to justify the use of Principal Component Analysis (PCA) to identify dietary patterns has been included in the introductory section (Line 69-81, text in red). 

“Dietary intake during pregnancy is commonly assessed through the intakes of energy, macronutrients, micronutrients or food groups [12,13], which may then be examined in relation to an imbalanced maternal diet and poor maternal nutritional status. Dietary pattern is a relatively new approach that describes a combination of commonly consumed foods [14] that allows for the diet to be described as a whole [15]. Nutritional health outcomes are often the result of multiple synergies among nutrients and foods rather than just the sum of the individual food [16]. Although there are several approaches to identifying dietary patterns of pregnant women, the posterior-approach derived from principal component analysis (PCA) is the most commonly used for deriving dietary patterns during pregnancy [10-12]. PCA is a technique to reduce a large of correlated variables into a smaller number of components [21,22], revealing the underlying structure within diets of the population. Numerous studies examining the role of PCA derived DPs in adverse pregnancy outcomes, such as GDM, pre-eclampsia, and pre-term birth [23–25] have been published.”

2. The authors presented conflicting findings about the significance of diet and dietary pattern for GDM. At some point they stated that DP2 and DP5 generated by PCA are significant predictors of GDM while latter they argued that dietary pattern as assessed by FFQ has no association with GDM. 

Response:

The association between dietary pattern and GDM was examined using logistic regression. Three models were examined. The first model adjusts for clinic and gestational week at OGTT. Model 2 adjusts for covariates of model 1 plus maternal age, and ethnicity. Model 3 then adjusts for covariates in model 2, plus income, pre-pregnancy BMI, medical history of GDM and family history of DM. This study found a significant association between DP and GDM in model 1 and model 2. However, after adjusting for covariates in model 3, the association becomes non-significant. The models are intended to portray the influence of different types of covariates (model 1 – clinic factors; model 2 – biological factors; model 3 – familial / environmental factors) on the association between DP and GDM. However, as suggested by reviewer, we will only retain 1 model, but which include the following covariates – clinic + biological factors + family history of DM + previous GDM.

3. In the abstract it is stated that “Our FFQ analysis did not reveal any clear associations between (pre)pregnancy dietary patterns and the risk to develop GDM (or not) …..” however in the results section, no result is provided regarding the direct association between frequency of consumption of food groups and risk of GDM.

Response:

We have revised the conclusion of the abstract accordingly (Line 47-53, text in red). 

Abstract

Overall, a relative low percentage of women with GDM was found in this cohort. The risk was lower in women with high adherence to a relatively unhealthy dietary pattern, i.e. DP 2 and DP 5. The lower body mass index (BMI) status and energy intake of women showing a high adherence to DP 2 in the first trimester may underlie the observed association with a lower GDM risk. Additionally, genetic variance might explain the less susceptibility to GDM despite HA to unhealthy DPs among non-GDM women. 

4. I recommend the authors to focus on the relationship between frequency of consumption of specific food groups (as measured by FFQ) and occurrence of GDM. Kind of multivariable analyses for assessing the association between the 17 food groups and GDM may provide practically meaningful findings.

Response:

We would like to retain the dietary pattern analysis as to emphasize the importance of food / nutrient interaction that represents usual diet. A paragraph to justify the usage of dietary pattern has been included in the Introduction section (refer to Response 1). 

5. Table 3 presents the results of three different models. I recommend the authors to make interpretation based on one model which they consider as the best.

Response:

We have revised Table 3 accordingly. 

6. The authors did not satisfactorily explain how DPs relatively higher in loadings for sugar, spread & creamer, sauces, condiments & spices, and oils & fats are associated with reduced risk of GDM. In the discussion section, they mentioned that the significant association was only observed among underweight and normal weight women; however, the results of this stratified analysis had not been present in the results section.

Response:

A lower BMI status and energy intake of women with high HA to DP 5 may underlie the observed association with a lower GDM risk. The findings of the stratified analysis have been included in the methods and results section. Please refer to Table 4 for the stratified analysis. 

Methods:

A stratified analysis was performed to determine the associations between DP 2 and DP 5 with GDM by pre-pregnancy BMI (Table 4).

Results:

The findings of the stratified analysis showed that the significant association between DP 5 and GDM was only observed among underweight or normal-weight women. Meanwhile, a significant association between DP 2 and GDM risk was found only among obese women.

Background

1. Please provide clear definitions for dietary pattern and “prudent diet”.

Response:

The definitions for the two terms have been incorporated:

Dietary pattern – (Line 72-73, text in red)

Prudent dietary pattern – (Line 96-100, text in red)

2. Page 3, line 73-77: the sentences are not clear. Please revisit them.

It could also be attributed to the different food items reported for the respective DPs. For example, the “prudent” DP in a prospective cohort study of women in Northern China was characterized by higher intakes of dark-coloured vegetables and deep-sea fish [12], whereas the “prudent” DP of women in the Born in Guangzhou Cohort Study (BIGCS) was predominantly dairy products, nuts, eggs, fish, soups, and fruits [13].

Response:

We have revised the sentences. (Line 96-100, text in red)

It could also be attributed to the different food items reported for the respective DPs. For example, the “prudent” DP, which is a diet delivers health benefits, observed in a prospective cohort study of women in Northern China was characterized by higher intakes of dark-colored vegetables and deep-sea fish [26], while the “prudent” DP of women in the Born in Guangzhou Cohort Study (BIGCS) was predominantly dairy products, nuts, eggs, fish, soups, and fruits [27]. 

2. The background section should give a highlight of what other studies conducted in Western or Asian population have found about the relationship between dietary pattern and GDM.

Response:

A paragraph on the relationship between dietary pattern and GDM has been included in the Introduction section. (line 83-89, text in red)

Associations between DP before and during pregnancy with risk of GDM has been examined [26–30]. Most studies showed that an adoption of the prudent pattern, diet that was rich in vegetables, fruits, whole grains and legumes showed significant lower risk of GDM [26–28], whereas adhering to the Western pattern, which is characterized by high intakes of red meat, processed meats and refined foods, was associated with increased risk of GDM [23,26,31]. There are several suggested mechanisms by which Western pattern may increase the risk of GDM through inflammation, and placental dysfunction [26,29,30].

Methods

3. Please comment on the adequacy of the sample size (power) for addressing the research questions at hand.

Response:

The adequacy of the sample size of pregnant women was checked to ensure a 5% statistical significance level and 80% power. (Line 126-128, text in red) 

For PCA analysis, the Kaiser-Meyer-Olkin measure of sampling adequacy was used to ensure that the sample size was adequate for these analyses. The Kaiser-Meyer-Oklin values were 0.68 (pre-pregnancy), 0.71 (first trimester), and 0.65 (second trimester). (line 160-163, text in red)

4. Line 184: please provide additional information on how the sensitivity analysis was made.

Response:

We have revised the sensitivity analysis. Sensitivity analyses were run between acceptable reporters and under-reporters of energy intake. Parity and total energy intake have been taken out from the sensitivity analysis model, as parity was not significant associated with GDM risk. For energy intake, the relationships between DP and GDM are likely to be at least partially mediated by total energy intake. As energy intake is likely to be on the causal pathway between dietary variables and the outcome of interest, it may be inappropriate to adjust for total energy when modelling DPs and GDM. However, we have to some degree, accounted for total energy intakes by identifying the dietary patterns: a higher z-score for the observed dietary pattern indicates that an individual has a high intake of food relevant to the specific dietary pattern. 

5. Line 188-91: not clear. Please revisit it.

Multivariable-adjusted odds ratio (OR) with 95% confidence interval (CI) were presented. Potential covariates included in the multivariable model were age (continuous), ethnicity (Malays/ Non-Malays), household income (continuous), pre-pre-pregnancy BMI (continuous), history of GDM (yes or no), and family history of DM (yes or no). In sensitivity analyses, additional adjustment for parity (yes or no), energy intake at that trimester (continuous), and omitting under-reporters (yes or no) was investigated. Repeated measures ANOVA was used to plot the DP trajectory with similar pattern by maternal glycemia categories. Three major patterns were identified before and during pregnancy: pattern 1 consisted of DP 1, DP 4, and DP 7, pattern 2 consisted of DP 2, DP 5, and DP 8, and pattern 3 included DP 3, DP 6 and DP 9. Significant level of the statistical analysis was set at the cut-off point of p< 0.05.

Response:

We have revised lines 188-91 accordingly. (Line 211-219, text in red)

Adjusted odds ratio (OR) with 95% confidence interval (CI) of the association between DPs and GDM were presented. A further stratified analysis was performed to determine the associations between DP 2 and DP 5 with GDM by pre-pregnancy BMI (Table 4). Repeated measures ANOVA was used to plot the DP trajectory for GDM and non-GDM groups. Three major patterns were identified at each time point: pattern 1 (DP 1, DP 4, and DP 7), pattern 2 (DP 2, DP 5, and DP 8), and pattern 3 (DP 3, DP 6 and DP 9). Sensitivity analyses were run between acceptable reporters and under-reporters of energy intake. The findings did not change the main findings (S1 Table). The cut off point for statistical significance was set at p<0.05.

6. Line 138-40: “The factor scores for each dietary pattern and each individual were calculated by summing intakes of food items weighted by their factor loadings”. Line 187-188: ANOVA was used to plot the DP trajectory with similar pattern by maternal glycemia categories. These sentences are not clear, please revisit them.

Response:

Line 164-166: The factor scores for each DP and for each individual item were calculated by summing the food item intakes weighted by their factor loadings.

Line 214-215: Repeated measures ANOVA was used to plot the DP trajectory for GDM and non-GDM groups.

7.“Three major patterns were identified before and during pregnancy: pattern 1 consisted of DP 1, DP 4, and DP 7, pattern 2 consisted of DP 2, DP 5, and DP 8, and pattern 3 included DP 3, DP 6 and DP 9.” The sentence is not clear.

Response:

Line 215-216: Three major DPs were identified at each time point: pattern 1 (DP 1, DP 4, and DP 7) pattern 2 (DP 2, DP 5, and DP 8) and pattern 3 (DP 3, DP 6 and DP 9). 

Results

8. Table 1: What was the justification for using age of 30 as a cut-off here? Use of 35 years as a cut-off would be practically meaningful b/c the risk of many obstetric complications tends to increase after the age of 35 years. Similarly, the cutoffs used for classifying height has not been justified.

Response:

We have revised the age group to 35 years. The classification of height was based on the tertile of height. 

9.Table 1: STPM/ Matric??? Not clear

Response:

STPM refers to the Malaysian Higher School Certificate and Matric refers to Matriculation. 

Footnote has been added for the abbreviation of STPM/Matriculation (Table 1).

10. If the information was collected during pregnancy, how did you manage to calculate pre-pregnancy BMI?

Response:

Women were requested to recall pre-pregnancy body weight. Pre-pregnancy body mass index (BMI) (kg/m2) was calculated as pre-pregnancy weight divided by the square of height and classified according to the recommendation of World Health Organization [14]. (Line 191-194, text in red)

11. Line 214-15: “Three specific DPs were observed at each time period before and during pregnancy”. It is not clear how you decided to extract three components at each round. Did you evaluate the scree plot or decide take all the components having Eigenvalues above 1? Or did you simply take the first three components with the highest Eigenvalues?

Response:

The number of factors was determined by eigen values greater than 1.5 [15]. (line 156-157, text in red).

12. Table 2: the combined total variance explained by the three principal components is relatively low (<40%) in all of the three rounds (pre-pregnancy, first trimester and second trimester). It is usually recommended that at least 60% of the variance should be explained by the extracted components. Do you think that the PCA has adequately explained the original 17 food groups?

Response:

We agree that the total variance explained by the 3 patterns at each time point is relatively low. However, several studies that used PCA to derive DP during pregnancy reported a comparable percentage of total variance explained (<30%) % [23,27,31]. We believe that it is reasonable to expect a total variance of < 40% in this study. We have addressed this issue as a study limitation. 

Limitation:

In this study, the total variance of 3 patterns explained in each time point is relatively low (<40%). However, this finding was comparable with other previous studies that reported a total variance of 16.6 – 32.9% [23,27,31].

13. I could not get the results of the “sensitivity analysis” mentioned in the methods section.

Response:

The result of the sensitivity analysis has been added. (S1 Table)

14. Line 259-271: the paragraph is not clear. Please revisit it again.

Differences in DP trajectory from pre-pregnancy until the second trimester of pregnancy between non-GDM and GDM women are depicted in Fig 2. While non-GDM women maintained high adherence to all three DPs before and during pregnancy, GDM women clearly seemed to change their diets over the course of pregnancy. For pattern 1 (DP 1, DP 4 and DP 7 – prudent diet, high in fruits, vegetables, nuts, seeds, legumes, eggs, milk and dairy), GDM positive women had low DP scores before pregnancy, decreased scores at the first trimester further decreasing at second trimester time point. For pattern 2 (DP 2, DP 5 and DP 8 – high in condiments & spices, and sugar, spread & creamer), a V-shaped trend was observed in GDM women, where DP score was at the lowest point in the first trimester and higher at the second trimester. For pattern 3 (DP 3, DP 6 and DP 9 – high protein, sugar and energy), GDM women showed an inverse V-shaped trend in that they had low DP score before pregnancy, the highest DP score at the first trimester and the lowest DP score at the second trimester.

Response:

We have revised the paragraph accordingly. (Line 298-309, text in red)

DP trajectory from pre-pregnancy to second trimester of pregnancy for non-GDM and GDM women are depicted in Figure 2. While non-GDM women maintained high adherence to all three DPs before and during pregnancy, GDM women changed their diets during pregnancy. For pattern 1 (DP 1, DP 4 and DP 7 – prudent diet, high in fruits, vegetables, nuts, seeds, legumes, eggs, milk and dairy), GDM women had low DP score before pregnancy, decreased score at the first trimester and further decreased score at second trimester. For pattern 2 (DP 2, DP 5 and DP 8 – high in condiments & spices, and sugar, spread & creamer), a V-shaped trend was observed in GDM women, where DP score was at the lowest point in the first trimester and higher at the second trimester. For pattern 3 (DP 3, DP 6 and DP 9 – high protein, sugar and energy), GDM women showed an inverse V-shaped trend in that they had low DP score before pregnancy, the highest DP score at the first trimester and the lowest DP score at the second trimester.

Discussion

15. Line 297-9: “Subsequently, a stratified analysis was performed to determine the associations between DP 2 and DP 5 with GDM by pre-pregnancy BM”. Please provide the findings of the stratified analysis in the results section. I see nothing about this analysis in the methods and results section.

Response:

Please refer to Table 4 for the stratified analysis. The findings of the stratified analysis have been included in the methods and results section as below. 

Thank you for asking me to review this interesting paper which has results contrary to expectations. The findings suggest that an unhealthy diet is protective against the development of gestational diabetes in a Malaysian population. The work is overall well conducted and well put together but the conclusions need further explanation and discussion in order to reconcile the findings. 

Response:

We have revised the conclusion accordingly. (Line 404-415, text in red)

Conclusion: 

The present study found that diets rich in sugar, spread and creamer, spices and condiments (DP 2 and DP 5) were significantly associated with reduced risk of GDM. The unexpected findings could be due to factors such as, lower BMI, and reduced energy intake. Furthermore, non-GDM women maintained a high DP score for all DPs from pre-pregnancy until second trimester of pregnancy, meanwhile GDM women showed lower and/or inconsistent DP scores, suggesting that non-GDM might have a lower genetic susceptibility to GDM as they are not at risk of GDM despite having also high DP scores for less healthy DPs (pattern 2 and pattern 3). A future large-scale prospective study or a well-designed randomized controlled trial is warranted to further confirm the association between the dietary pattern and the GDM in the Malaysian population. Such findings could inform efforts to strengthen existing or develop appropriate health and nutrition strategies to address the increasing rate of GDM in Malaysia.

Women eating DP 2 and 5 had a diet rich in sugar, spread and creamer, spices and condiments, but this is unlikely to account for all calories. It would be helpful to know if these women were eating much else besides? Perhaps a 'typical day' idea of what this diet looked like would be useful. 

Response:

The detail of dietary pattern before and during pregnancy has been published elsewhere (DOI: 10.4162/nrp.2019.13.3.230). (Line 168-170, text in red)

It would also be helpful to know what exactly the number of calories were for each group. If this group was eating only small amounts of other food stuffs then it is possible the calorie amount was very different. What %carbohydrate was seen in each group? Is there information on dietary macronutrients? It is also possible that the difference in carb or fat content also drives this association.

Response:

Yes, we have run the analysis as indicated (energy and macronutrients with GDM). However, only total energy showed a significant association. Please refer to supplementary tables 2. 

I personally would urge caution in interpreting these findings and would suggest that the relevance of this to public health remains unclear. This is not reason to reject this paper, but there is a risk the findings could be misinterpreted.

Response:

Thank you for your valuable comments. We have revised the manuscript accordingly. I hope the revised manuscript will meet the requirements of PlosOne. 

The study findings should be interpreted with caution as this pattern was relatively higher in loadings for sugar, spread & creamer, sauces, condiments & spices, and oils & fats. (Line 316-318, text in red).

---

## [Decision Letter · Decision Letter 1]

17 Dec 2019

The association between dietary patterns before and in early pregnancy and the risk of gestational diabetes mellitus (GDM): data from the Malaysian SECOST cohort

PONE-D-19-22261R1

Dear Dr. Yong,

We are pleased to inform you that your manuscript has been judged scientifically suitable for publication and will be formally accepted for publication once it complies with all outstanding technical requirements.

With kind regards,

Samson Gebremedhin, PhD

Academic Editor

PLOS ONE

Additional Editor Comments:

Please accommodate the additional comments provided by reviewer 2.

Reviewers' comments:

Reviewer's Responses to Questions

**Comments to the Author**

1. If the authors have adequately addressed your comments raised in a previous round of review and you feel that this manuscript is now acceptable for publication, you may indicate that here to bypass the “Comments to the Author” section, enter your conflict of interest statement in the “Confidential to Editor” section, and submit your "Accept" recommendation.

Reviewer #2: All comments have been addressed

2. Is the manuscript technically sound, and do the data support the conclusions?

Reviewer #2: Yes

3. Has the statistical analysis been performed appropriately and rigorously? 

Reviewer #2: Yes

4. Have the authors made all data underlying the findings in their manuscript fully available?

Reviewer #2: No

5. Is the manuscript presented in an intelligible fashion and written in standard English?

Reviewer #2: Yes

6. Review Comments to the Author

Reviewer #2: This version is overall much improved. A few minor points:

Abstract:

HA should be defined (last line)

Methods:

line 126. Why were women excluded?

Can you be sure that women with pre-existing glucose intolerance were excluded? If not, then you need to acknowledge this as a limitation.

lines 299-310 - good to remember that these are based on relatively small numbers of women. What are the related confidence intervals? Are these changes significant?

Is there any information on total calories consumed? If so this should be added to table 2.

7. PLOS authors have the option to publish the peer review history of their article (what does this mean?). If published, this will include your full peer review and any attached files.

Reviewer #2: No

---

## [Editor Report · Acceptance letter]

23 Dec 2019

PONE-D-19-22261R1 

The association between dietary patterns before and in early pregnancy and the risk of gestational diabetes mellitus (GDM): data from the Malaysian SECOST cohort 

Dear Dr. Yong:

I am pleased to inform you that your manuscript has been deemed suitable for publication in PLOS ONE. Congratulations! Your manuscript is now with our production department. 

With kind regards,

on behalf of

Dr. Samson Gebremedhin 

Academic Editor

PLOS ONE